# Evaluation of Novel Guanidino-Containing Isonipecotamide Inhibitors of Blood Coagulation Factors against SARS-CoV-2 Virus Infection

**DOI:** 10.3390/v14081730

**Published:** 2022-08-05

**Authors:** Flavio De Maio, Mariagrazia Rullo, Modesto de Candia, Rosa Purgatorio, Gianfranco Lopopolo, Giulia Santarelli, Valentina Palmieri, Massimiliano Papi, Gabriella Elia, Erica De Candia, Maurizio Sanguinetti, Cosimo Damiano Altomare

**Affiliations:** 1Dipartimento di Scienze di Laboratorio e Infettivologiche, Fondazione Policlinico Universitario A. Gemelli IRCCS, I-00168 Rome, Italy; flavio.demaio@unicatt.it (F.D.M.); giulia.santarelli22@gmail.com (G.S.); vplabcemi@gmail.com (V.P.); massimiliano.papi@unicatt.it (M.P.); 2Department of Pharmacy–Pharmaceutical Sciences, University of Bari Aldo Moro, I-70125 Bari, Italy; mariagrazia.rullo@uniba.it (M.R.); modesto.decandia@uniba.it (M.d.C.); rosa.purgatorio@uniba.it (R.P.); gianfranco.lopopolo@newchemspa.it (G.L.); 3Department of Veterinary Medicine, University of Bari Aldo Moro, I-70125 Bari, Italy; gabriella.elia@uniba.it; 4Department of Translational Medicine and Surgery, Catholic University of Rome, I-00168 Rome, Italy; erica.decandia@unicatt.it

**Keywords:** factor Xa, thrombin, anticoagulants, SARS-CoV-2, COVID-19

## Abstract

Coagulation factor Xa (fXa) and thrombin (thr) are widely expressed in pulmonary tissues, where they may catalyze, together with the transmembrane serine protease 2 (TMPRSS2), the coronaviruses spike protein (SP) cleavage and activation, thus enhancing the SP binding to ACE2 and cell infection. In this study, we evaluate in vitro the ability of approved (i.e., dabigatran and rivaroxaban) and newly synthesized isonipecotamide-based reversible inhibitors of fXa/thr (cmpds **1**–**3**) to hinder the SARS-CoV-2 infectivity of VERO cells. Nafamostat, which is a guanidine/amidine antithrombin and antiplasmin agent, disclosed as a covalent inhibitor of TMPRSS2, was also evaluated. While dabigatran and rivaroxaban at 100 μM concentration did not show any effect on SARS-CoV-2 infection, the virus preincubation with new guanidino-containing fXa-selective inhibitors **1** and **3** did decrease viral infectivity of VERO cells at subtoxic doses. When the cells were pre-incubated with **3**, a reversible nanomolar inhibitor of fXa (*K*_i_ = 15 nM) showing the best in silico docking score toward TMPRSS2 (pdb 7MEQ), the SARS-CoV-2 infectivity was completely inhibited at 100 μM (*p* < 0.0001), where the cytopathic effect was just about 10%. The inhibitory effects of **3** on SARS-CoV-2 infection was evident (ca. 30%) at lower concentrations (3–50 μM). The covalent TMPRSS2 and the selective inhibitor nafamostat mesylate, although showing some effect (15–20% inhibition), did not achieve statistically significant activity against SARS-CoV-2 infection in the whole range of test concentrations (3–100 μM). These findings suggest that direct inhibitors of the main serine proteases of the blood coagulation cascade may have potential in SARS-CoV-2 drug discovery. Furthermore, they prove that basic amidino-containing fXa inhibitors with a higher docking score towards TMPRSS2 may be considered hits for optimizing novel small molecules protecting guest cells from SARS-CoV-2 infection.

## 1. Introduction

Coronavirus disease 2019 (COVID-19) caused by severe acute respiratory syndrome coronavirus 2 (SARS-CoV-2) was first detected in China (December 2019) [1], before becoming a global pandemic emergency, which affected more than 400 million people and caused about six million deaths over the world. Besides the development of vaccines, unprecedented efforts are ongoing worldwide for disclosing new druggable targets and novel or repurposed drugs to fight the COVID-19 pandemic [2,3,4], which led to the identification of the first orally active drug, paxlovid [5,6].

COVID-19 clinical symptoms include fever, dry cough, sore throat, dyspnea, headache, and interstitial pneumonia, which can potentially progress to alveolar damage and consequent respiratory failure [7]. COVID-19 high mortality rate is not only correlated to viral replication in lung epithelial cells, but also mediated by a strong dysregulated host immune response defined as “cytokine storm syndrome” [8].

COVID-19 infection increases the risk of arterial and venous thrombosis, leading to considerable interest in antithrombotic treatment to prevent and treat these COVID-19 related complications. In addition, endothelial injury causing microvascular pulmonary thrombosis is associated with poor clinical outcomes in patients with interstitial pneumonia. In this context, some studies focused on patients already on oral anticoagulant (OAC) therapy when diagnosed with COVID-19, based on the hypothesis that they could be at lower risk of adverse outcomes as compared to nontreated patients. Conflicting results on the role of OAC therapy on diverse clinical outcomes of COVID-19 patients were reported. Some studies showed that prior use of therapeutic anticoagulation did not improve survival in hospitalized COVID-19 patients and that similar outcomes were observed both for patients treated with vitamin K antagonists (VKAs) or direct-acting oral anticoagulants (DOACs) [9,10]. In contrast, a retrospective cohort study reported that COVID-19 patients on OAC treatment at the time of infection and during the disease showed significantly lower risk of all-cause mortality at 21 days [11]. Another study reported that, among elderly patients hospitalized for COVID-19, OAC therapy had a significantly lower mortality rate than patients not receiving anticoagulation [12,13]. A retrospective observational study from CORIST registry also showed that OAC may have protective effects on adverse outcomes by COVID-19 in hospitalized patients with atrial fibrillation and that fewer adverse events occurred in patients on DOACs compared to patients on VKAs [14]. Interestingly, another study reported that high-risk atrial fibrillation patients on OAC therapy had a lower risk of receiving a positive COVID-19 test and severe COVID-19 outcomes [15].

Preliminary data suggest that oral anticoagulants (OAC), in addition to exerting their antithrombotic activity to prevent early microthrombotic events and venous thromboembolism, might also provide protection against infection by SARS-CoV-2. To this regard, a previous experimental study showed that direct factor Xa (fXa) inhibitors may prevent SARS-CoV from entering into human cells by inhibiting the spike protein (SP) cleavage by factor Xa into the S1 and S2 subunits [16].

SARS-CoV-2 internalization into the host cells of lung epithelia is mediated by the interaction of the viral SP with angiotensin converting enzyme II (ACE2) and the cleavage of SP protein by the transmembrane protease serine 2 (TMPRSS2), a plasma-membrane-anchored serine protease playing a role in promoting viral uptake and fusion at the cell membrane of several human respiratory viruses [17]. Recent data suggest that polymorphisms of TMPRSS2 may modulate the severity of SARS-CoV-2 infection [18]. Other enzymes, including the blood coagulation proteases fXa and thrombin (thr) can be synthesized under pathological conditions in several tissues, and in pulmonary tissues as well, and could mediate the cleavage of viral SP, thus representing other targets for antiviral drugs [19,20]. Intriguingly, the efficacy of nafamostat mesylate, which is a guanidine/amidine antithrombin and antiplasmin agent, has been recently reported as a covalent inhibitor of TMPRSS2 [21].

Herein, we focused on investigating two FDA-approved (i.e., dabigatran and rivaroxaban) and three novel reversible inhibitors of thr/fXa (Figure 1, cmpds **1**–**3**), synthesized by Lopopolo et al. [22], for their effects on SARS-CoV-2 infectivity of VERO cells (African green monkey kidney epithelial cells) [23,24]. Nafamostat mesylate, the anti-fXa/thr activity of which has also been determined herein, was used as a positive control in the virus infection assays.

## 2. Materials and Methods

**Chemicals.** Nafamostat mesylate, dabigatran, and rivaroxaban were purchased from CliniSciences (Nanterre, France). Compounds **1**, **2**, and **3** were synthesized by Dr. Modesto de Candia in the laboratory of Department of Pharmacy-Pharmaceutical Sciences, University of Bari Aldo Moro (Bari, Italy), according to previously reported methods [22,25,26,27,28,29,30,31]. The purity of the compounds **1**–**3** was ascertained by elemental analyses (C, H, and N), performed on a Euro EA3000 analyzer (Eurovector, Milan, Italy), by the Analytical Laboratory Service of the Department. The results were in agreement to within 0.4% of theoretical values. All the tested compounds showed higher than 95% purity. Water used for the preparations of solutions was of 18.2 MV (Milli-RiOs/QA10 grade, Millipore Corp., Bedford, MA, USA). All the other chemicals and solvents used were of analytical grade and were purchased from Sigma-Aldrich, Milan (Italy), Alfa Aesar, and Thermo-Fischer GmbH, Kandel (Germany).

**Enzyme inhibition assays.** The time-dependent inhibition of fXa and thrombin was assessed by determining the hydrolysis rates of the selective synthetic chromogenic substrates, which were monitored at 405 nm. Enzymes and substrates used were (final concentrations): 0.41 unit·mL^−1^ for bovine thrombin (Sigma-Aldrich, Milan, Italy) and 50 μM S-2238 (d-Phe-Pip-Arg-*p*-NA) from Chromogenix AB-Instrumentation Laboratories (Milan, Italy); 4 nM recombinant human factor Xa and 0.04 μM S-2765 (Z-d-Arg-Gly-Arg-*p*-NA) from Chromogenix AB-Instrumentation Laboratories (Milan, Italy). The enzyme solutions (100 μL) were mixed with 2 μL of DMSO solution containing the test compound or DMSO alone as a control and incubated at different times (5, 15, and 30 min). Reactions were initiated by adding 100 μL of substrate solutions, and the increase in absorbance was monitored for 10 min. The initial velocities were determined and the concentrations of the inhibitors required to reduce the control velocity by 50% (IC_50_) were determined by a sigmoidal regression. Each kinetic was repeated three times. After determination of the IC_50_ values, the inhibition constants (*K*_i_) were calculated by applying the Cheng–Prousoff equation.

**Stability in water and human plasma**. Stability of nafamostat and compound **3** was determined either in phosphate buffer solution (pH 7.4) or human plasma at 37 °C [28,32]. To study the stability in the aqueous medium, 100 μL of a 10 mM stock solution of tested compound in DMSO was diluted to a 10 mL volume by adding pH 7.4 PBS solution (0.05 M phosphate buffer pH 7.4 containing 0.15 M KCl) to obtain a solution at the final concentration of 100 μM. This solution was incubated at 37 ± 0.5 °C and, at previously established time intervals, 100 μL of sample was withdrawn and analyzed by RP-HPLC on a 1260 Infinity Quaternary LC system (Agilent Technologies, Milan, Italy), equipped with an autosampler and a photodiode array detector. A Phenomenex Luna C8 column 5 μm (150 × 3.0 mm i.d.) was used as the stationary phase. The analytes were eluted in isocratic by mixing in different ratios a 10 mM ammonium formate (AF) aqueous solution (pH 4.5) and acetonitrile (ACN), at a constant flow rate of 0.4 mL/min, by injecting 5 μL of sample solutions. The mobile phase composition was 85% AF/15% ACN for nafamostat analysis, and 55% AF/45% ACN for compound **3** analysis.

The pseudo-first-order rate constants (*k*_obs_) for the hydrolysis were calculated from the slopes of the reported linear plots of log (% of remaining compound) against time. Each experiment has been performed in triplicate. For the stability measurements in pooled human serum, 10 μL of a 10 mM stock solution of each compound in DMSO was added to 1 mL of pooled defrosted human serum, preheated at 37 ± 0.5 °C. The obtained solutions were incubated at 37 ± 0.5 °C (final compound concentration: 100 μM). At various times, aliquots of 100 μL of the serum solution were taken and deproteinized by mixing with 400 μL of cold ACN. The suspension was vortexed for 1 min and centrifuged 10 min at 3500 rcf. The supernatant was filtered (4 mm PTFE filters, pore size 0.2 µm) and 10 μL of the filtered solution was analyzed by RP-HPLC, as reported above.

**Molecular modeling.** Docking simulations were performed using the crystal structure of nafamostat in complex with human TMPRSS2 (entry code 7MEQ) retrieved from the Protein Data Bank [29]. The protein pretreatment in order to remove water molecules, assign bond orders, add hydrogen atoms, create disulfide bonds, fill in missing side chains and loops using Prime, and cap termini was carried out with the Protein Preparation Wizard available from Schrödinger. Protonation states at pH 7.0 ± 2.0 and tautomers for histidine residues were predicted with the Epik function, and default parameters were used for the optimization of hydrogen-bond assignment. A restrained energy minimization step only on hydrogens was finally applied to the protein using the OPLS3e force field. The ligand preparation was performed using the LigPrep module in the Schrödinger Suite v.2019-1. Compounds **1**–**3** were initially drafted in the Maestro panel; then protonation states were calculated at pH 7.0 ± 2.0, and up to a maximum of 32 stereoisomers could be computed for each ligand by generating all possible combinations using the OPLS3e force field. The “Receptor Grid Generation” panel of Glide was used to generate the grid files with grid points calculated within an enclosing box of 20 Å defined by the X-ray co-ordinates of nafamostat taken from 7MEQ [30]. Docking calculations were performed for each ligand using the Glide module of Schrödinger, using standard (SP) mode and default settings (van der Waals radius scaling parameters: scaling factor of 0.80 and partial charge cut-off of 0.15; dock flexibly; add Epik state penalties to docking score; perform post-docking minimization). The scoring function of Glide known as “Docking score” was used for estimating the binding affinity. Pictures were elaborated with the Schrödinger suite.

**Cytotoxicity Assay.** The cytotoxicity on confluent monolayers of VERO cells (African green monkey kidney cell line) has been determined in 96-well plates by treating cells with serial two-fold dilutions of each test compound (ranging from 1.5 to 200 µM) in a final volume of 100 µL. The plates were incubated at 37 °C. Untreated cells served as a control, whilst three wells for each drug concentration were filled with complete culture medium D-MEM, without cells, and used as a blank. Four wells were used for each drug concentration and controls, respectively. All the experiments were carried out in duplicate.

After 72 h of incubation, all plates were examined microscopically to assess general morphology of cells. The occurrence of cell rounding was recorded in the wells containing five serial drug dilutions starting from the maximum concentration of 200 µM for all compounds. Then, 70 µL of XTT (Invitrogen-Thermo Fischer Scientific, Rodano (MI), Italy) was added to each well and plates were incubated at 37 °C for an additional 4 h to allow the production of formazan. Optical densities were determined at 450 nm against a reference wavelength at 650 nm by using an automatic spectrophotometer (microtitre plate reader, Biorad 550). Percentage of cytotoxicity was calculated by applying the following formula:
(1)
$$\%\;cytotoxicity=\left(\frac{OD\;control\;cells-OD\;treated\;cells}{OD\;control\;cells}\right)\times 100$$


Results were finally expressed as CC_50_, representing the drug concentration at which the viability of treated VERO cells decreased to 50% of control cells.

**Antiviral activity assays. Cell culture and SARS-CoV-2 infection.** African green monkey kidney (VERO) epithelial cells (ATCC CCL-81) were cultured in Dulbecco modified Eagle’s medium (DMEM) supplemented with 10% inactivated fetal calf serum (FCS) (Euroclone, Milan, Italy), 1 mM glutamine (Euroclone, Milan, Italy), and 1% streptomycin–penicillin antibiotics (Euroclone, Milan, Italy) and incubated in humidified atmosphere (5% CO_2_ at 37 °C), as reported elsewhere [31]. Cells were washed with sterile warm phosphate buffer (PBS), trypsinized, and counted. The monolayer was obtained by seeding cells in 48-well plates (Nest) at a final concentration of 7 × 10^4^ cell/mL. Infection was carried out with SARS-CoV-2 when >90% confluent monolayer was reached, approximately 1 × 10^6^ cell/well after 72 h.

Briefly, cells were washed with sterile warm PBS and then infected with 0.1 mL of solution containing SARS-CoV-2. Cells were incubated for two hours at standard atmosphere conditions (37 °C, 5% of CO_2_) until infection solution was removed, and new fresh DMEM medium (supplemented with 2% FCS, 1 mM glutamine, and 1% streptomycin–penicillin antibiotics) was added. Cells were incubated as previously described and infection status was evaluated daily. All the experiments that involved SARS-CoV-2 manipulation were carried out in Biosafety level 3 laboratory (BSL3) in the Institute of Microbiology of IRCCS–Fondazione Policlinico Gemelli.

**Measurement of SARS-CoV-2 infection inhibition by DOACs.** Antiviral activity against SARS-CoV-2 has been assessed for a set of commercially available compounds (camostat mesylate, nafamostat mesylate, dabigatran, argatroban monohydrate, and rivaroxaban), along with a selection of previously homemade synthesized inhibitors of the coagulative serine proteases **1**–**3**, by applying two experimental settings.

*Assay A (preincubation test)*. Each compound was diluted in culture medium, then added to confluent monolayers of VERO cells at a final concentration of 100 μM. After 2 h incubation at 37 °C, the compounds were removed, the cells were washed with sterile warm phosphate buffer (PBS), and, finally, were infected with a suspension of SARS-CoV-2 (≈10^5^ virus particles/mL), as previously described.

*Assay B (co-incubation test)*: to determine the direct effect of compounds on viral particles, SARS-CoV-2 suspension was mixed into each compound solution and incubated at 37 °C before performing VERO cells infection.

For both experimental settings, viral suspension was removed and replaced with new fresh culture media two hours later. Cells were monitored daily and cytopathic effect was evaluated at 72 h after infection.

**Cell viability and assessment of viral load.** Crystal violet staining was performed to evaluate cell viability and cellular disruption following the SARS-CoV-2 infection and its replication. Cells were fixed by using 4% paraformaldehyde (Sigma-Aldrich, Burlington, MA, USA) for 30 min and then stained by using Crystal violet (Sigma-Aldrich, USA) for 30 min. After incubation, five washes were carried out and images were acquired by using Cytation instrument (Cytation Cell Imaging Reader-Agilent BioTek, Santa Clara, CA, USA). Immunofluorescence assay was performed to assess viral replication. Briefly, cells were fixed as previously described. After three washes, fixed cells were permeabilized and a blocking step was performed by using PBS supplemented with 0.3% bovine serum albumin (BSA) (Sigma-Aldrich, USA). Monoclonal rabbit anti-Spike S1 subunit was used (Novusbio, clone CR3022) and the plate was incubated 3 h at room temperature. After washes, secondary anti-rabbit IgG-FITC-labelled antibody was added (Invitrogen), and the signal was detected by using Cytation instrument.

**Image analysis.** Images were analyzed using the freely available ImageJ version 1.47v (NIH, USA). Every set of tiff images corresponding to crystal violet staining was analyzed through the “Process > Batch > Macro tool”. Each image was converted to an 8-bit image. Minimum and maximum thresholds were manually set for each batch of images to correctly convert areas to white and black, respectively. Prior to performing the “Measure” tool of ImageJ, images were processed with the “Smooth” and “Convert to Mask”. The fraction of the area covered by cells was then automatically stored in the results file. IF images were generated by merging an image of cells acquired with direct light and the corresponding image acquired with 476 nm light (UV) by ImageJ software;s built-in tool “Image > Color > Merge” using the grey and green channels, respectively.

**Statistical analysis.** All experiments were replicated at least three times. Graphpad Prism software ver. 9 was used to collect and to analyze the data. All data were represented as box plots and analyzed by nested one-way ANOVA comparison tests, followed by Dunnett’s correction.

## 3. Results and Discussion

Compounds **1**–**3** are isonipecotanilide derivatives which differ for bearing appended at N1 the amidino group (**1** and **3**) or the isopropyl group (**2**), and for *ortho*- (**1** and **2**) or *meta*-substitution (**3**) of the 5-(5-chlorothiophen-2-yl)isoxazol-3-yl)methoxy moiety with respect to the carboxamide functionality. The inhibition constant (*K*_i_) values (Table 1) show that the isonipecotamide derivatives **1**–**3** are all highly potent fXa-selective inhibitors, with a (sub)nanomolar potency (*K*_i_ = 0.3–15 nm) close to that of rivaroxaban *K*_i_ = 5 nm. In our assay conditions, nafamostat (NAF) proved to be a quite selective thr inhibitor, with a *K*_i_ of 10 nM close to that achieved by dabigatran. Regarding the inhibition mechanism, the isonipecotamide-based inhibitors **1**–**3**, dabigatran and rivaroxaban as well all proved to be reversible inhibitors of blood coagulation proteases. In contrast, nafamostat, similarly to the behavior observed by others against TMPRSS2, proved to act as a covalent inhibitor of thr and fXa (time-dependent inhibition data summarized in Appendix A).

The inhibition potency (*K*_i_) of NAF, in contrast with that of compound **3**, was strictly dependent upon the incubation time of the inhibitor prior to adding the excess of the competing substrate (see Appendix A). For instance, tested without preincubation, nafamostat resulted in a good thr inhibitor, with a *K*_i_ value of 77 nM. Preincubated with α-thrombin for 15 and 30 min, nafamostat resulted in a stronger inhibitor, with *K*_i_ equal to 13 and 5 nM, respectively. In a parallel experiment, the isonipecotamide derivative **3** did not show any significant time-dependent change in inhibition constant.

The stability of nafamostat and **3** was determined both in PBS at pH 7.4 and in human serum, monitoring their hydrolytic degradation by HPLC according to previously reported methods [28,32]. Regarding nafamostat, in the literature, there is a lack of information about its stability in buffered solutions, whilst it has been reported that it undergoes hydrolysis to a large extent in whole blood (more rapid) and plasma [33].

In our study (Figure 2), nafamostat was stable enough in PBS at pH 7.4 (half-life about 45 h), whereas it was hydrolyzed in human pooled blood serum in a very short time (half-life 16 min), which can negatively affect its in vivo efficacy. In contrast, the fXa-selective inhibitor **3** (data in Appendix A) was stable both in PBS pH 7.4 and pooled blood human serum (hydrolytic degradation less than 10% after 24 h).

Cell toxicity of each tested compound was preliminarily evaluated in mock-infected VERO cells with a spectrophotometric formazan-based cell viability assay. Cytotoxicity data (Table 1) were expressed as % of nonviable cells at the highest concentration used (100 µM) and the 50% cytotoxic concentration (CC_50_). All the compounds showed CC_50_ values exceeding 200 µM, with the only exception of **3** (CC_50_ = 155 µM), which, at 100 µM, exhibited low cytotoxicity (about 12%).

Considering the high sequence homology between the examined blood coagulation serine proteases and TMPRSS2, we wanted to assess in silico the potential of our compounds to bind and inhibit this enzyme involved in SP cleavage, that favors the virus entry into the guest cell. TMPRSS2 protein belongs to the S1A serine protease family, characterized by a conserved fold, with an Arg binding site and three catalytic residues (Asp, His, and Ser) in close proximity to the Arg binding site. TMPRSS2 and other proteases of the S1A family, such as the coagulation factor Xa, plasminogen, and plasma kallikrein, share at least 80% structural identity in the region close to the active site identified as S1-S1’. Ligands capable of binding S1-S1’ subsites might also inhibit TMPRSS2.

To test our hypothesis, molecular docking calculations were carried out using the flexible ligand docking program of Schrödinger’s Glide module [34] with SP docking score function. Compounds **1**–**3** were docked into the binding site of NAF using the X-ray crystal structure of human TMPRSS2, in which the catalytic Ser441 is acylated by 4-guanidinobenzoyl moiety of NAF (PDB entry: 7MEQ; Figure 3A). The X-ray structure of the covalent complex shows that the guanidino group forms a salt bridge with the side chain of Asp435 reinforced by a network of H bonds (HBs) with the backbone CO of Gly464 and Ser436. Two other HB contacts between the backbone NHs of Gly439 and Ser 441 and the benzoyl CO further contribute to stabilize the complex.

The reversible guanidino-containing inhibitors **1** and **3** (Figure 3B and Figure 3D, respectively), in their highest scored docking poses, may form, with their amidino group appended at N1, electrostatic and HB interactions with the residues Asp435, Ser436, and Gly464, which are very similar to those engaged by the NHC(=NH)NH_2_ group in the acyl moiety of the covalent inhibitor NAF. The *meta*-substituted isomer **3**, which showed a slightly better docking score compared with the *ortho* isomer **1**, might achieve additional face-to-edge π–π interactions between Trp461 residue and the thiophene ring (Figure 1E). The *ortho*-substituted isonipecotanilide **1** (Figure 3B) seems to bind TMPRSS2 in a less efficient docking pose, likely due to the loss of hydrophobic and aromatic interactions with Trp461. The visual inspection of the binding mode of the *ortho*-substituted **2** (Figure 3C), in which the guanidino group is replaced by the isopropyl group, suggests that the lowest docking score in the small ligand series may likely be due to lack of suitable electrostatic interactions, HBs, and hydrophobic/aromatic interactions. The basic 1-isopropyl-piperidine ring is directed out of the binding pocket and the bi(hetero)aryl group does not form any contact with aromatic/hydrophobic counterparts.

On this basis, the potential anti-infectious activity against SARS-CoV-2 of all the investigated compounds was assessed in VERO cell line by applying two different experimental settings: preincubation (Figure 4A) and coincubation (Figure 4C) with DOACs. Indeed, the VERO cells express a TMPRSS2 isoform sharing about 80% of structure homology with the human protein, also retaining higher similarity in the active binding site [35,36]. Each compound was first tested at one-point concentration of 100 μM, incubating it for two hours before infection with SARS-CoV-2. Untreated and uninfected cells were used as controls. Viral infectivity was measured 72 h after infection, when cells were fixed and stained by using crystal violet [37]. Pretreatment with fXa/thr inhibitors did not show any reduced cytophatic effect of the cellular monolayer.

Conversely, compound **3** showed statistically significant protective effects against SARS-CoV-2 at a noncytotoxic dose. A weak (nonsignificant) protective effect was observed with the isonipecotamide derivative **1** and NAF but not with **2**, dabigatran, and rivaroxaban, which did not show any ability to reduce the virus infectivity. Compounds **1** and **3** were able to reduce SARS-CoV-2 infectivity to the value of untreated cells (statistical significance was observed only for **3**).

In the second experimental setting, each compound was coincubated with the SARS-CoV-2 infection solution before infecting VERO cells (Figure 3C). While a trend similar to that of the pretreatment experimental setting was observed, compound **3** appeared as the best candidate against the viral infection (Figure 4D), suggesting a possible role of the cellular TMPRSS2 and the membrane-associated fXa protein.

In Figure 5A, representative images of crystal violet staining were reported for the isonipecotamide derivative **3** and nafamostat mesylate. Additionally, immunofluorescence assay was performed to corroborate the ability of compound **3** to reduce SARS-CoV-2 entry. Crystal violet staining, measuring cell viability, correlated with low fluorescence levels, which indicated the reduced viral entry and replication after treatment with compound **3** (Figure 5B,C). Conversely, the covalent TMPRSS2 and thrombin inhibitor nafamostat, although showing some effect in reducing viral entry (around 20% inhibition), did not achieve statistically significant activity against SARS-CoV-2 infection in terms of cell viability (Figure 5B,C). Nafamostat mesylate appeared to preserve around 60% of the cellular monolayer, with SARS-CoV-2 detected in ~25% of the cells (Figure 5B,C).

To corroborate these findings, nafamostat mesylate and the isonipecotamide derivative **3** were preincubated with VERO cells at scalar concentrations (from 100 to 3 μM) before infection, and both cell viability and viral replication were measured (see Appendix A in Appendix A). Albeit no scalar activity was detected, compound **3** confirmed previous results showing 100% inhibition of virus infectivity at the highest test concentration; however, inhibitory effects of compound **3** on SARS-CoV-2 infection were evident (~30%) at lower concentrations (3–50 µM).

Conversely, the covalent TMPRSS2 and thrombin inhibitor, nafamostat, was confirmed not to have statistically significant activity against SARS-CoV-2 infection in the whole range of scalar concentrations tested (3–100 µM).

Interestingly, crystal violet staining evidenced a slightly different cell morphology between uninfected VERO cells and cells treated with the compound **3** and then infected. Although cell viability appeared similar, as previously described, VERO cells treated with isonipecotamide derivative **3** showed a less defined outline membrane (Figure 5A).

Given the chemical structures of the compounds investigated herein, which resemble cationic amphiphilic drugs (CADs), we cannot exclude the incidence of membrane phenomena such as phospholipidosis at 100 µM, the only concentration in which the basic guanidino derivative 3 exerts a significant protective effect against the virus infectivity. Phospholipidosis may explain why dose dependency is not observed for **3** at concentrations lower than 100 µM. Drug-induced phospholipidosis, which is a lysosomal storage disorder characterized by excessive accumulation of phospholipids, may represent a real limitation in drug discovery of anti-COVID-19 agents. As reported by Tummino and coll. [38], the antiviral activity of CADs may be a consequence of phospholipidosis on lysosome functions. If CADs induce an alteration in lysosome permeability, the affected ion channels may block the viral entry. This hypothesis is consistent with the antiviral activity of tamoxifen, which proved to disrupt calcium homeostasis [39]. How much CAD-induced phospholipidosis is a factor of cytoprotection against virus or cytotoxicity is matter of scientific discussion. In principle, we cannot exclude that our guanidino-containing compounds may induce phospholipidosis. However, looking at the physicochemical properties of a number of antiviral CADs, it has been inferred that molecules exceeding threshold values of p*K*_a_ (>7.4) and clogP (>3) could be more likely to induce phospholipidosis. In the small molecular set examined herein, all the compounds are basic (p*K*_a_ >> 7.4), whereas, except the less active amino derivative **2** (clogP = 4.02), the other compounds, namely nafamostat (2.28) and the guanidino derivatives **1** (1.84) and **3** (2.43), do not exceed the threshold clogP value of 3 and may have a lower risk of inducing phospholipidosis.

## 4. Conclusions

These findings highlighted the ability of guanidino-containing isonipecotamide derivatives, such as compound **3**, to reduce SARS-CoV-2 entry in VERO cells, even though other possible effects inside the cells cannot be excluded. The isonipecotamide derivative **3**, which showed, in vitro, a low nanomolar inhibition potency toward fXa and, in silico, a good propensity to bind (and likely to inhibit) TMPRSS2, proved herein to totally block SARS-CoV-2 entry at high concentration (100 μM), maintaining both cellular integrity and low viral load. We cannot rule out that compound **3**, as a cationic amphiphilic compound, could induce phosphilipidosis, which could even explain the absence of a dose–response relationship in the protection from viral infection and/or cytotoxicity. However, its clogP under the threshold value of **3** may indicate a lower risk of phospholipidosis induction. While further structure-based molecular optimization of the hit molecule **3** will be aimed at improving activity and decreasing intrinsic cell toxicity, these findings provide evidence that some basic chemotypes of DOACs, such as guanidino and amidino derivatives, may have potential in SARS-CoV-2 drug discovery.

## Figures and Tables

**Figure 1 viruses-14-01730-f001:**
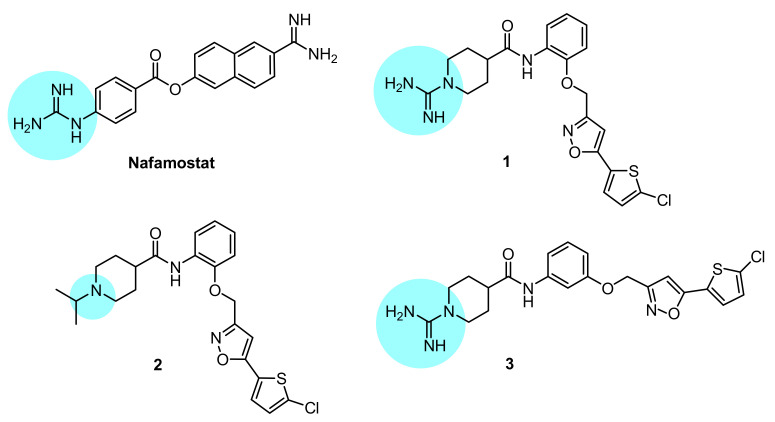
Chemical structures of the compounds investigated in this study; the basic anchoring point of each structure is highlighted with a light blue backcloth.

**Figure 2 viruses-14-01730-f002:**
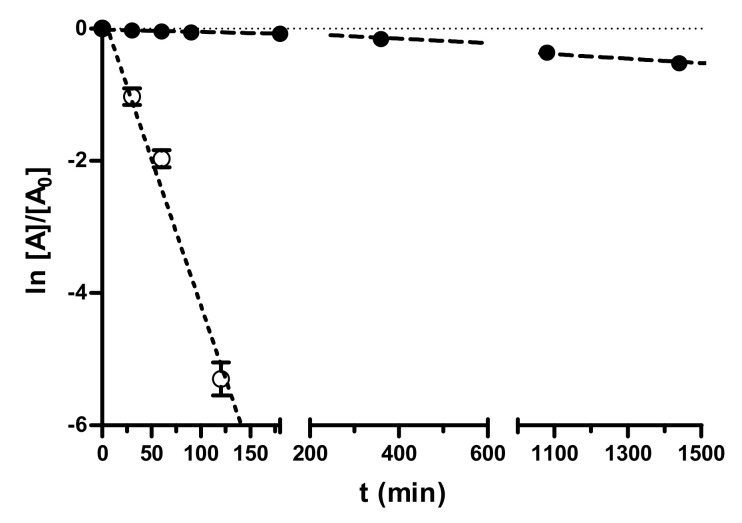
Pseudo-first-order kinetic plots of the decomposition of nafamostat in 0.05 M PBS (pH 7.4, 0.15 M KCl) and in pooled human serum at 37 °C. Data points represent means ± SD of three independent measurements by RP-HPLC at various time points in 24 h monitoring.

**Figure 3 viruses-14-01730-f003:**
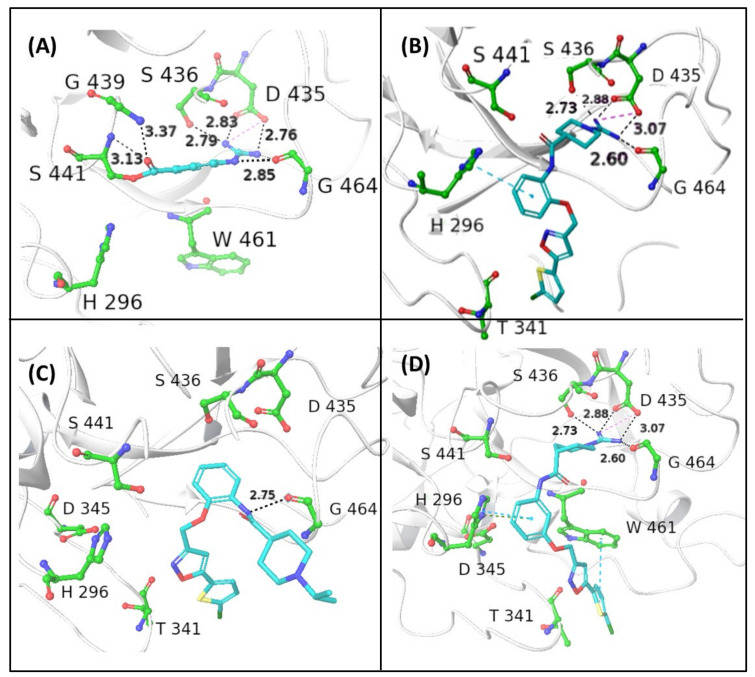
In silico prediction of binding modes of the investigated compounds as ligands of type 2 transmembrane serine protease (TMPRSS2). (**A**) Crystal structure of human TMPRSS2 in complex with the covalent inhibitor nafamostat (pdb code: 7MEQ); the binding interactions with the surrounding residues of the catalytic S441 acylated with 4-guanidinobenzoyl moiety are shown. In silico Glide top-scored docking poses of the novel inhibitors **1** (**B**), **2** (**C**), and **3** (**D**) into human TMPRSS2. The inhibitors are represented in sticks, while the amino acid residues are shown in ball-and-sticks. Hydrogen bonds (HBs) and π–π interactions are shown as black and blue dashed lines, respectively (distances in Å); docking scores are expressed in kcal/mol: **1** (−7.242), **2** (−5.852), and **3** (−7.474).

**Figure 4 viruses-14-01730-f004:**
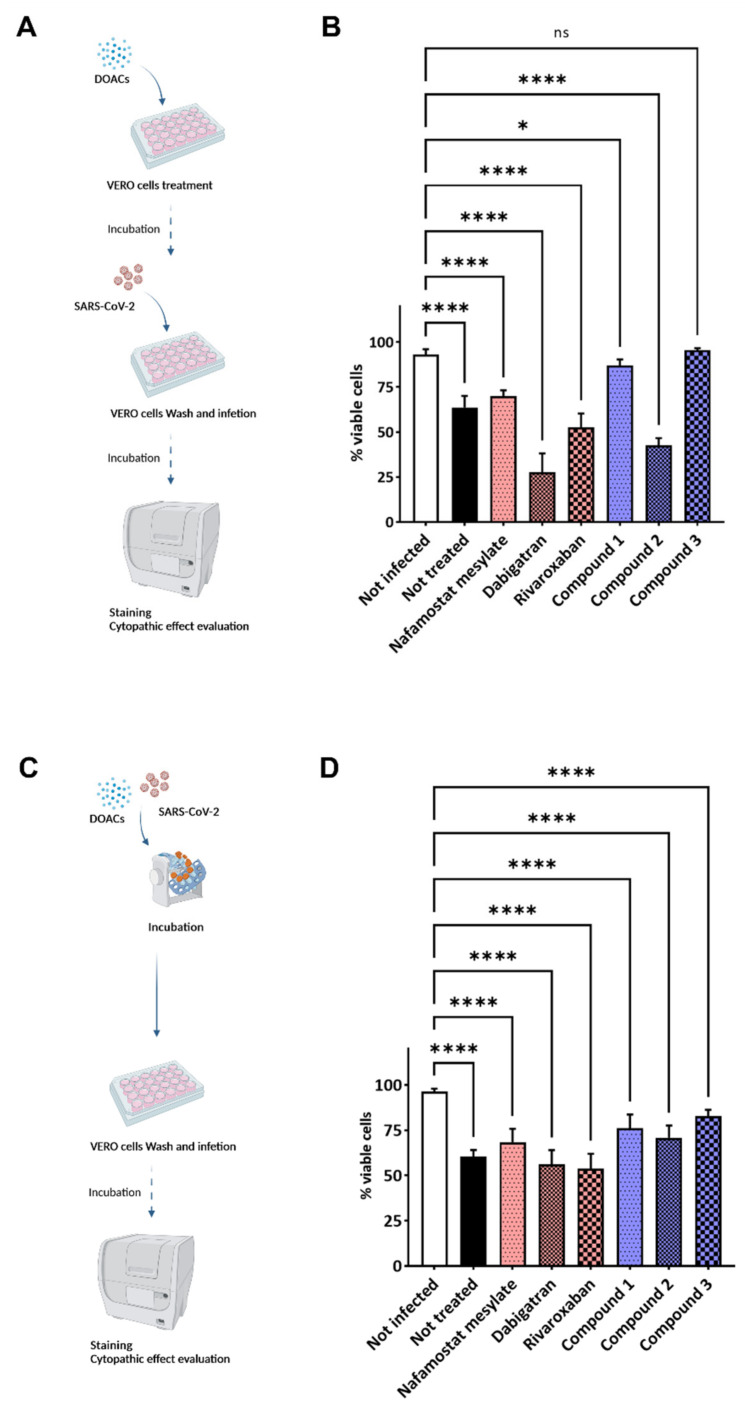
Schematic representations of the experimental settings and results. (**A**,**B**) Preincubation experimental setting: VERO cells were treated with the selected compounds diluted in cell culture medium at final concentration of 100 μM for 2 before infection with SARS-CoV-2. (**C**,**D**) Coincubation experimental setting: a suspension of SARS-CoV-2 previously incubated with the selected molecules, at final concentration of 100 μM, was used to infect VERO cells. Untreated cells were used as infection control. Infection solution was removed after two hours, and new fresh sterile medium was added. Cells were daily monitored until monolayer was fixed before staining with crystal violet. Cytation instrument was used to acquire images of each well and analysis was performed by ImageJ software measuring the integrity of the cellular monolayer. Bar plots, showing average and standard deviation, summarize the result of repeated experiments (panels **B** and **D** highlight the results obtained for the experimental setting represented in **A** and **C**, respectively). Data were analyzed by using nested one-way ANOVA comparison test, followed by Dunnett’s correction. Statistically significant results are indicated according to *p* value as follows: * *p* < 0.05; **** *p* < 0.0001 (each sample was compared to uninfected control).

**Figure 5 viruses-14-01730-f005:**
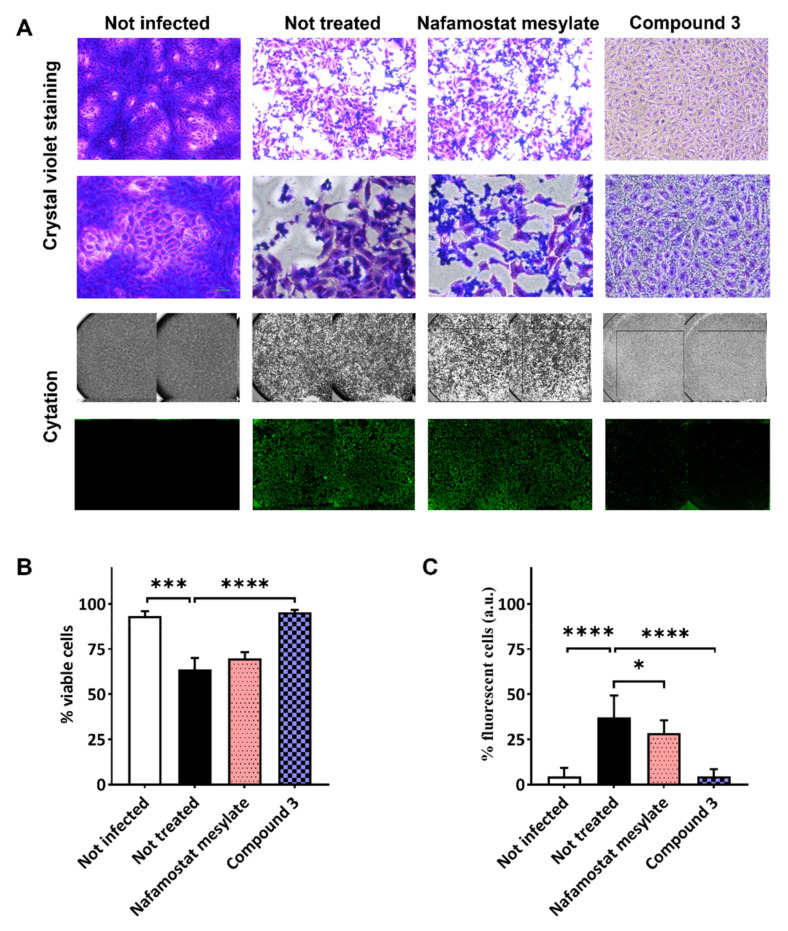
SARS-CoV-2 infection assay in VERO cells after treatment with nafamostat mesylate and compound **3**. VERO cells were treated with the test compounds diluted in cell culture medium at final concentration of 100 μM. Two hours later, cells were washed and infected with SARS-CoV-2 clinical isolate. Untreated cells were used as infection control. Cells were monitored daily until monolayer was fixed before staining with crystal violet or immunofluorescence. Cytation instrument was used to acquire images of each well. (**A**) Representative images of crystal violet staining (10× and 20× in the first and second lanes, respectively) and an example of images acquired by Cytation instrument in visible light to analyze cytopathic effect/cell viability (third lane), and in immunofluorescence to evaluate infected cells (fourth lane). Bar plots showing average and SD of repeated experiments following crystal violet staining (**B**) and immunofluorescence (**C**). Statistical significance was assessed by using nested one-way ANOVA comparison tests, followed by Dunnett’s correction, indicated according to *p* value as follows: * *p* < 0.05; *** *p* < 0.001; **** *p* < 0.0001.

**Table 1 viruses-14-01730-t001:** In vitro affinities (*K*_i_ values) to thrombin and factor Xa, and cytotoxicity data (Vero E6 cell line).

Compounds	Enzyme Inhibition, *K*_i_ (nM) ^a^		VERO Cells Toxicity ^b^	
	Factor Xa	Thrombin	Cytotoxicity at 100 µM	CC_50_ (µM)
**Nafamostat**	600	10	16.0 ± 0.8	>200
**1**	2	6820	20.3 ± 0.9	>200
**2**	0.3	11,000	7.45 ± 0.24	>200
**3**	15	15,000	11.6 ± 0.2	155 ± 16
**Dabigatran**	5100	4.2	18.3 ± 0.8	>200
**Rivaroxaban**	5	12,000	29.4 ± 0.8	>200

^a^ Inhibition constants (*K*_i_ values, nM) of the blood coagulation factors thrombin (thr) and activated factor X (fXa); dabigatran and rivaroxaban, as well-known thr-selective and fXa-selective inhibitors, respectively, were used as positive controls. The *K*_i_ values were calculated by applying the Cheng–Prousoff equation to IC_50_ values by regression (GraphPad Prism software ver. 5.01); data are means of three independent measurements (SD < 5%); ^b^ cytotoxicity data on VERO cell line are means of three independent measurements ± SD.

## Data Availability

Not applicable.

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
