# Peer review of "Evaluation of Novel Guanidino-Containing Isonipecotamide Inhibitors of Blood Coagulation Factors against SARS-CoV-2 Virus Infection"

_viruses, 2022, doi:10.3390/v14081730_

Round 1

Reviewer 1 Report

In this article De Maio et al. evaluate potential SARS-CoV-2 antivirals targeting host factors that are required for cleavage and activation of the spike protein prior to binding and infection. The authors provide a clear and concise rationale as part of a wider introduction as to why the research was performed and how it may be of impact. The material and methods provide a detailed and reproducible description of the methodology.  The authors investigate enzyme inhibition in comparison to known inhibitors, identifying that they are potent against fXa and much less so thrombin, alongside in-silico prediction of docking to TMPRSS2, a protease essential to SARS-CoV-2 infectivity. These compounds are further tested in cell culture models to determine effectiveness of SARS-CoV-2 inhibition, with compound 3 suggested to be a potent inhibitor of infection. Whilst this finding is novel, I have several concerns and minor comments that require addressing before this can be considered for publication and these are outlined below.

Major comments

·         Drug-induced phospholipidosis has been identified as a major barrier to drug discovery for SARS-CoV-2, summarised by Tummino et al. 2021, Science. Given the chemical structure of the compounds tested in this screen, assays should be performed to determine whether phospholipidosis is a contributing factor to observed results across a range of dilutions, which may provide insights into why dose-dependency is not observed for compound 3 below 100 µM. This is not necessarily important for the main body of text but should be at least included in supplementary materials and referred to.

·         Figure 4 B,D,E+G – a significant difference in % viable cells is observed in the not treated sample when comparing B/D with E/G. Why is this? Given that dabigatran and compound 2 in 4B have a similar viability to E+G not treated control this may suggest that the not treated control in B is potentially unreliable? I appreciate that cross-comparing is not best practice, however an approx. 40% difference in viability across a comparable assay is concerning.

·         Figure 4 – statistical analysis should be performed and displayed for all samples not just a select few. If all unlabelled bars are non-significant this could alternatively be mentioned in the materials and methods and figure legend.

·         As a suggestion figure 4 could be broken down into two smaller figures – A-D and E-H. Addition of representative images to accompany graphical representation could also be considered, as it would make key points clearer for the reader – eg when looking at fluorescent cells – but is not necessary.

·         Figure 2 – it is not clear what each dataset on the graph represents and requires a key. There are only 2 visible datasets, however in the text nafamostat and compound 3 are discussed in to conditions and I therefore believe the latter data has not been presented?

Minor comments

·         Line 36 – E6 VERO cells should be changed to VERO E6 cells for consistency.

·         Line 47 -remove ‘infectious’ from COVID-19 abbreviation. COVID-19 = coronavirus disease 2019.

·         Line 53 – missing full stop.

·         Line 54 – inflammatory response (in particular the aggressive cytokine storm) is a major issue in disease development and should be included in this paragraph when mentioning alveolar damage etc.

·         Line 67 – provide full names of acronyms VKA and DOAC.

·         Line 74 and onwards – assuming AF is disease related, atrial fibrillation? Please make clear. AF is also used as the abbreviation for ammonium formate (line 140), if this is different please change to make clearer.

·         Line 108 – 22 of 20-22 references appears to be in superscript. Change to normal text.

·         Line 138 – assuming the µ is supposed to be µm? Add in if so.

·         Line 194 – antiviral activity assays – include which strain of SARS-CoV-2 is used.

·         Line 217 and figure 4A – it is not clear whether inhibitors are removed at the point of infection or removed. Please clarify.

·         Line 260 – text states nafamostat has a thrombin Ki of 10 nM, but the table says 5. Change accordingly as to which is incorrect.

·         Line 261-264 – Reword this sentence as it’s difficult to understand in the current form.

·         Figure 3 – Increase text size for residues and distances as it is hard to see due to size and blurriness.

·         Line 350 – change ‘damage’ to something more appropriate. Eg cytopathic effect or CPE.

·         Line 354 – authors state ‘compounds 3 and 1 were able to reduce SARS-CoV-2 infectivity to value of not treated cells’. I’m assuming this is meant to be compared to not infected?

·         Figure 4 – better align all graphs. A+C writing is too small on the schematic to read clearly. A-D and E-H use two different font sizes, make the same.

·         Figure 4 – add x-axis labels to make the graphs easier to interpret. Particularly for E-H.

·         Figure 4 E-H – highlight in the legend that these datasets were pre-incubation assays, as defined by the authors.

·         Figure 4 F+H – the authors may consider a smaller y-axis range, given that nothing is observed above 50%. This would aide in visualising differences between concentrations.

·         Figure 4 legend (lines 363 and 366) – Add E6 to VERO cells to make consistent with rest of document.

Author Response

In this article De Maio et al. evaluate potential SARS-CoV-2 antivirals targeting host factors that are required for cleavage and activation of the spike protein prior to binding and infection. The authors provide a clear and concise rationale as part of a wider introduction as to why the research was performed and how it may be of impact. The material and methods provide a detailed and reproducible description of the methodology. The authors investigate enzyme inhibition in comparison to known inhibitors, identifying that they are potent against fXa and much less to thrombin, alongside in-silico prediction of docking to TMPRSS2, a protease essential to SARS-CoV-2 infectivity. These compounds are further tested in cell culture models to determine effectiveness of SARS-CoV-2 inhibition, with compound 3 suggested to be a potent inhibitor of infection. Whilst this finding is novel, I have several concerns and minor comments that require addressing before this can be considered for publication and these are outlined below.

Answer: We thank the reviewer for the overall positive comments. In the following point-by-point reply, we address the specific points raised by the reviewer.

Major comments

Drug-induced phospholipidosis has been identified as a major barrier to drug discovery for SARS-CoV-2, summarized by Tummino et al. 2021, Science. Given the chemical structure of the compounds tested in this screen, assays should be performed to determine whether phospholipidosis is a contributing factor to observed results across a range of dilutions, which may provide insights into why dose-dependency is not observed for compound 3 below 100 µM. This is not necessarily important for the main body of text but should be at least included in supplementary materials and referred to.

Answer: Thanks to the referee for her/his comment. Drug-induced phospholipidosis, that is a lysosomal storage disorder characterized by excessive accumulation of phospholipids, may be a hurdle in identifying new compounds or repurposing old drugs as anti-covid agents. Indeed, as argued by Tummino and coll. (Science 2021), the antiviral activity of cationic amphiphilic drugs (CADs) may be a consequence of phospholipidosis on lysosome functions. CADs have a hydrophilic amine head that can be protonated in the endolysosomal compartment. If CADs induce an alteration in lysosome permeability through phospholipidosis, the affected ion channels may block the viral entry. This hypothesis is consistent with the mechanism of action of tamoxifen, for example, which exerts its antiviral activity by disrupting calcium homeostasis (B. Breiden & K. Sandhoff, Biol. Chem. 2020). However, how much phospholipidosis is a factor of cytoprotection from virus entry or of cytopathy is matter of scientific debate.

In principle, we cannot exclude that our novel guanidino-containing compounds may induce phospholipidosis. Nevertheless, Tummino et al., based on physicochemical properties of the examined drugs, suggested that CADs exceeding the threshold values of basicity (pKa > 7.4) and lipophilicity (clogP > 3) could more likely induce phospholipidosis. In our small molecular series, all the examined compounds are basic (pKa > 7.4), whereas excepted the less active amino derivative 2 (clogP = 4.02), the other compounds, namely the benzamidine derivative nafamostat (2.28), and the novel guanidino derivatives 1 (1.84) and 3 (2.43), do not exceed the threshold clogP value of 3, and may be compounds with lower risk of phospholipidosis.

We’ll assess such a risk factor for molecules resulting from the ongoing optimization studies, whereas herein we show a new updated Figure 4 and introduced some comments in the discussion section related to the physicochemical risk factors likely related to phospholipidosis possibly related to the observed results.

Figure 4 B,D,E+G – a significant difference in % viable cells is observed in the not treated sample when comparing B/D with E/G. Why is this? Given that dabigatran and compound 2 in 4B have a similar viability to E+G not treated control this may suggest that the not treated control in B is potentially unreliable? I appreciate that cross-comparing is not best practice, however an approx. 40% difference in viability across a comparable assay is concerning.

Answer: We thank the reviewer for the observation, and we apologize for the mistake, but some values were erroneously excluded during final analysis. We have now corrected in the revised figure 4. The slight differences in B/D panels of the old figure 4 (positive control viability of 63.7±13.0 and 60.3±11.2) and E/G panels of the old figure 4 (positive control viability of 40.2±17.0 and 39.0±18.0), did not result significant when compared and are due to experimental procedures involving these biological systems. On the other side, these experimental settings were sequential: panels B/D corresponded to screening results of different molecules (we have previously demonstrated how this method can be used to measure cell viability after infection [De Maio 2021]) so that a slight, but not significant, differences in preliminary experiments (i.e. panels B/D of the old figure 4) increased stringency in selecting promising compounds to be further assayed (panels E/G of the old figure 4). However, in the revised version we decided to maintain only panels A-B and C-D, displacing in the Supplementary Data file (Figure S1) the other panels showing the lack of dose-response relationships for the most active compound 3 as compared with nafamostat mesylate (NAF).

Figure 4 – statistical analysis should be performed and displayed for all samples not just a select few. If all unlabeled bars are non-significant this could alternatively be mentioned in the materials and methods and figure legend.

Answer: Statistical analysis has been repeated and corrected if any. Missing information has been now reported in the figure legend and experimental section.

As a suggestion figure 4 could be broken down into two smaller figures – A-D and E-H. Addition of representative images to accompany graphical representation could also be considered, as it would make key points clearer for the reader – eg when looking at fluorescent cells – but is not necessary.

Answer: Complying with the above suggestion, we revised Figure 4 and displaced the panels pertaining the dose-response relationships for the compound 3 and nafamostat mesylate (NAF) into the Supplementary data file in Figure S1. Moreover, for the sake of clarity, a new Figure 5 has been added, which reports the view after crystal violet staining and immunofluorescence in Cytation instrument, as described in Materials and Methods.

Figure 2 – it is not clear what each dataset on the graph represents and requires a key. There are only 2 visible datasets, however in the text nafamostat and compound 3 are discussed into conditions and I therefore believe the latter data has not been presented?

Answer: Maybe, this figure might induce confusion. In Figure 2 we reported the hydrolytic degradation kinetics of nafamostat only, under two different conditions: buffered aqueous solution and human pooled serum, both at physiological pH 7.4. Moreover, as reported in the body of manuscript, under the same experimental conditions, compound 3 showed proved to be highly stable (its concentration is reduced by less than 10% in 24h). An additional table has been implemented now in the Supplementary data file (Table S2), which highlights the residual concentration of nafamostat and compound 3 at each time. The related sentences in the manuscript have been modified as follows: “In our study (Figure 2), nafamostat resulted stable enough in PBS at pH 7.4 (half-life about 45 h), whereas it was hydrolyzed in human pooled blood serum in very short time (half-life 16 min), which can negatively affect its in vivo efficacy. In contrast, the fXa-selective inhibitor 3 (data in Table S2 in Supplementary data file) resulted stable both in PBS at pH 7.4 and pooled blood human serum (hydrolytic degradation less than 10% after 24 h).”

Minor comments

Line 36 – E6 VERO cells should be changed to VERO E6 cells for consistency.

Answer: The changes have been done, and often we correctly wrote “VERO cells”.

Line 47 -remove ‘infectious’ from COVID-19 abbreviation. COVID-19 = coronavirus disease 2019.

Answer: We apologize for the mistake.

Line 53 – missing full stop.

Answer: Done.

Line 54 – inflammatory response (in particular the aggressive cytokine storm) is a major issue in disease development and should be included in this paragraph when mentioning alveolar damage etc.

Answer: Thank the reviewer for the suggestion. The information has been added in the revised manuscript.

Line 67 – provide full names of acronyms VKA and DOAC.

Answer: Done.

Line 74 and onwards – assuming AF is disease related, atrial fibrillation? Please make clear. AF is also used as the abbreviation for ammonium formate (line 140), if this is different, please change to make clearer.

Answer: Since atrial fibrillation appears a very few times in the main text, we can leave AF only as the abbreviation of ammonium formate.

Line 108 – 22 of 20-22 references appears to be in superscript. Change to normal text.

Answer: Done.

Line 138 – assuming the µ is supposed to be µm? Add in if so.

Answer: The referee is right. Correction done.

Line 217 and figure 4A – it is not clear whether inhibitors are removed at the point of infection or removed. Please clarify.

Answer: We thank the reviewer for the suggestion. This point has now better described.

Line 260 – text states nafamostat has a thrombin Ki of 10 nM, but the table says 5. Change accordingly as to which is incorrect.

Answer: The value in the table has been corrected to 10 nM in the Table.

Line 261-264 (now 267-270)– Reword this sentence as it’s difficult to understand in the current form.

Answer: The sentence has been reworded as following: “Regarding the inhibition mechanism, in contrast with the isonipecotamide-based inhibitors 1-3, as well as dabigatran and rivaroxaban, which all proved to be reversible inhibitors of blood coagulation proteases, nafamostat proved to act as covalent inhibitor of thr and fXa (Table S1A-B, Supplementary Data file).”

Figure 3 – Increase text size for residues and distances as it is hard to see due to size and blurriness.

Answer: Done.

Line 350 – change ‘damage’ to something more appropriate. Eg cytopathic effect or CPE.

Answer: We changed as suggested.

Line 354 – authors state ‘compounds 3 and 1 were able to reduce SARS-CoV-2 infectivity to value of not treated cells’. I’m assuming this is meant to be compared to not infected?

Answer: The reviewer is right. The statistical analysis has been checked and better introduced in Figures 4 and 5, as well as in Figure S1 in the Supplementary data file. Although both compounds 1 and 3 showed a positive trend in terms of reducing infectivity, only compound 3 revealed statistical significance.

Figure 4 – better align all graphs. A+C writing is too small on the schematic to read clearly. A-D and E-H use two different font sizes, make the same.

Figure 4 – add x-axis labels to make the graphs easier to interpret. Particularly for E-H.

Figure 4 E-H – highlight in the legend that these datasets were pre-incubation assays, as defined by the authors.

Figure 4 F+H – the authors may consider a smaller y-axis range, given that nothing is observed above 50%. This would aide in visualizing differences between concentrations.

Figure 4 legend (lines 363 and 366) – Add E6 to VERO cells to make consistent with rest of document.

Answer: Figure 4 has been revised and a new Figure 5 has been added; legends and other axes labels revised according to the referee’s remarks.

Reviewer 2 Report

The study of De Maio et al. is devoted to determination of the effects of three novel anti-protease inhibitors on VERO E6 cell survival to SARS-CoV-2 infection. The authors demonstrate that the proposed compounds could enhance the survival of cells. Altogether, the study is of interest and could be published in Viruses after text revision, however, the relevance of the possible usage of high doses of anticoagulants in SARS-CoV-2 is very disputable.

Points to be corrected: - The phrase " the blood coagulation proteases fXa and thrombin (thr) expressed in pulmonary tissues " (lines 90-91) should be corrected, as these proteins do not bind to live cell membranes and are synthethised only as non-active zymogens, and, therefore, they could not be expressed in any tissues. -The usage of VERO E6 cells as a single model of COVID-19 infection should be explicitly explained both in the Introduction and in Methods. Some references to previous studies comparing SARS-CoV-2 effect on monkey and human cells should be added to the Introduction.  -The authors use molecular docking to demonstrate the potential inhibition of the  TMPRSS2   by the proposed compounds. However, the experiments were conducted in VERO E6 cells - not human cells, and it should be noted that TMPRSS2 of VERO E6 cells has only 80% (428/533 aa) similarity to human  TMPRSS2. Some discussions should be added on this point.  -The authors demonstrated that the cells should be incubated with high concentrations of the compounds (100 uM). However, if one recalculates the compounds' affinities to the enzyme from the docking scores, one obtains 5.4(1), 57(2) and 3,8(3) uM - much lower values. Cound the difference come from the difference in molecular sequence, mentioned above, or from other possible action of the compounds on the cells? - Some Discussion of the obtained results should be added to the manuscript. The points to be discussed, apart from mentioned above, should include the viability of using anticoagulants with high affinity for active coagulation proteases as an anti-viral agens, and weather the obtained on cell culture results could be translated to the infection of an organism.

Author Response

The study of De Maio et al. is devoted to determination of the effects of three novel anti-protease inhibitors on VERO E6 cell survival to SARS-CoV-2 infection. The authors demonstrate that the proposed compounds could enhance the survival of cells. Altogether, the study is of interest and could be published in Viruses after text revision, however, the relevance of the possible usage of high doses of anticoagulants in SARS-CoV-2 is very disputable.

Answer: We thank the reviewer for the overall positive comments on the whole manuscript. We agree with the reviewer that high doses of anticoagulants could not represent a real antiviral treatment and that further studies of hit-to-lead optimization are needed. However, we showed the biological potential of these molecules in line with some clinical observations. In the point-by-point reply, we will address the specific points raised by the reviewer.

Points to be corrected:

The phrase " the blood coagulation proteases fXa and thrombin (thr) expressed in pulmonary tissues " (lines 90-91) should be corrected, as these proteins do not bind to live cell membranes and are synthesized only as non-active zymogens, and, therefore, they could not be expressed in any tissues.

Answer: We thank the referee for her/his suggestion. The sentences in the manuscript have been revised, as follows: “Other enzymes, including the blood coagulation proteases fXa and thrombin (thr) can be synthesized under pathological conditions in any tissue, expressed in pulmonary tissues as well, and could mediate the cleavage of viral SP, thus representing other targets for antiviral drugs.”

The usage of VERO E6 cells as a single model of COVID-19 infection should be explicitly explained both in the Introduction and in Methods. Some references to previous studies comparing SARS-CoV-2 effect on monkey and human cells should be added to the Introduction.

Answer: We thank the reviewer for the suggestion. We have now included some references on the use of VERO cells to study SARS-CoV2- infection in both introduction and materials and methods section. We have also discussed this approach in the discussion.

The authors use molecular docking to demonstrate the potential inhibition of the TMPRSS2 by the proposed compounds. However, the experiments were conducted in VERO E6 cells - not human cells, and it should be noted that TMPRSS2 of VERO E6 cells has only 80% (428/533 aa) similarity to human TMPRSS2. Some discussions should be added on this point.

Answer: Thanks to the referee for her/his constructive suggestion. Recombinant stable clone of Vero E6 cell line constitutively express the full length human TMPRSS2 serine proteases and represents the main applied experimental model to assess potential of selective inhibitors as anti-covid agents (Takayama, K. Trends in Pharmacological Sciences, 2020, 41, 513). Really, the expressed TMPRSS2 isoform in Vero E6 cell line share 80% of structural homology but, as reported elsewhere (see the new references 33 and 34), the active site retain a very high similarity the human isoform, thus justifying the use of VERO cell line. As suggested, we add to the text as follows: “On this basis, the potential anti-infectious activity against SARS-CoV-2 of all the studied compounds was assessed in VERO cell line by applying two different experimental settings: pre-incubation (Figure 4-A) and co-incubation (Figure 4-C) with DOACs. Indeed, the VERO cells express a TMPRSS2 isoform sharing about 80% of structural homology with the human protein, retaining high similarity in the active binding site.”

The authors demonstrated that the cells should be incubated with high concentrations of the compounds (100 mM). However, if one recalculates the compounds affinities to the enzyme from the docking scores, one obtains 5.4 (1), 57 (2) and 3.8 (3) mM - much lower values. Could the difference come from the difference in molecular sequence, mentioned above, or from other possible action of the compounds on the cells?

Answer: Actually, the docking score should be always taken only as a figure of merit of a given binding mode. No conclusion can be drawn about the prediction accuracy of the real biological data, even the enzymes’ inhibition constants. On the other hand, we cannot rule out other mechanisms cooperating to increase the inhibitory potency of compound 3 toward virus infectivity. Several text changes and addition in the revised manuscript should have better highlighted this point.

- Some Discussion of the obtained results should be added to the manuscript. The points to be discussed, apart from mentioned above, should include the viability of using anticoagulants with high affinity for active coagulation proteases as an anti-viral agents, and whether the obtained on cell culture results could be translated to the infection of an organism.

Answer: The main purpose of the study is investigating if novel potent guanidino-containing inhibitors of fXa/thr may have potential in treating SARS-CoV-2 infection. The data show that there is potential for these hit molecules, which now require further work aimed at better characterizing their mechanism(s) of action and carrying out a hit-to-lead optimization. With some text changes and revisions made the discussion of the results has been improved.

Round 2

Reviewer 1 Report

All comments have been suitably answered/addressed. The additional figure 5 contains really nice data that beautifully shows the inhibition of CPE by compound 3. I have no further comments to add, other than to wish the authors all the best in the future.